# Current Status of Liverwort Herbaria Specimens and Geographical Distribution in China

**DOI:** 10.3390/plants13182583

**Published:** 2024-09-15

**Authors:** Jiaqi Cui, Xiuhua Yang, Xiaoyu Li, Jitong Li, Siqi Dong, Hongfeng Wang, Chengjun Yang

**Affiliations:** 1Collage of Forestry, Northeast Forestry University, Harbin 150040, China; 2022120205@nefu.edu.cn (J.C.); yangxiuhua169329@163.com (X.Y.); xiaoyu_li2022@163.com (X.L.); lijitong101@163.com (J.L.); 18845932973@163.com (S.D.); 2Northeast Asia Biodiversity Research Center, Northeast Forestry University, Harbin 150040, China

**Keywords:** liverworts, specimen data, collection status, integrity assessment, climatic factors, geographical distribution

## Abstract

Specimen data play a crucial role in geographical distribution research. In this study, the collection information of liverwort specimens in China was compiled and analyzed to investigate the history, current status, and limitations of liverwort research in China. By utilizing the latest systematic research findings and corresponding environmental data, a niche model was developed to offer theoretical support for exploring the potential geographical distribution and diversity of liverwort resources. A total of 55,427 liverwort specimens were collected in China, resulting in the recording of 1212 species belonging to 169 genera and 63 families. However, there are imbalances in the distributions of liverwort data among different groups, collection units, and geographical areas, with families such as *Lejeuneaceae*, *Porellaceae*, and *Plagiochilaceae* having the highest number of specimens. Similarly, genera such as *Porella*, *Frullania*, and *Horikawaella* were well represented. Remarkably, 125 species had specimen counts exceeding 100. Unfortunately, approximately 51.77% of the species had fewer than 10 recorded specimens. There were four obvious peaks in the collection years of the bryophyte specimens in China, among which the largest collection occurred from 2010 to 2023. Notably, the number of specimens collected at different stages closely aligned with the history of taxonomic research on liverworts in China. The results of the integrity of the liverwort collection indicate that there is insufficient representation of some families and genera, with a concentration of common and widely distributed large families and genera. Tropical and subtropical humid areas are key regions for liverwort diversity, with water and temperature being the primary environmental factors influencing their geographical distribution. The specific temporal and spatial data of species recorded from plant specimens will enhance the study of species diversity, comprehensive protection, and sustainable utilization. Additionally, these data will contribute to the investigation of large-scale biodiversity distribution patterns and the impact of global change on diversity.

## 1. Introduction

Plant specimens are voucher materials collected and preserved by plant researchers during long-term scientific research activities. They reflect the human understanding of a certain group and are also physical records of natural heritage [1]. The digitization of specimens helps the main information, such as the habitat status, geographical distribution and collection of specimens, to be permanently preserved to the greatest extent possible, which greatly facilitates the sharing of specimen information [2]. In 2004, the African Plants Initiative (API) marked the commencement of digitizing global model specimens. The Latin American Plants Initiative (LAPI) and the Global Plants Initiative (GPI) were established. The digitization of plant specimens and the establishment of digital plant herbaria have been widely promoted worldwide [3]. A number of network platforms have been established around the world to provide data query and data sharing services, such as China’s National Specimen Resource Sharing Platform (NSII, http://www.nsii.org.cn/2017/home.php, accessed on 30 May 2024), the National Specimen Digital Platform of the United States (https://www.idigbio.org/, accessed on 12 July 2024), the Australian Biodiversity Information System (http://www.ala.org.au/, accessed on 13 July 2024), the Digital Plant Herbarium of Moscow State University of Russia (https://plant.depo.msu.ru/, accessed on 12 July 2024), and the Global Biodiversity Information Facility (GBIF (https://www.gbif.org/, accessed on 30 June 2024), which integrates the largest sharing platform for global specimen data from multiple locations and units.

In 2006, China’s plant specimens began to be fully digitized. By 2017, more than 80 herbaria (rooms) of teaching and research institutions in China had carried out the digitization of plant specimens [2]. After the digitization of plant specimens from major herbaria in China, they were mainly shared with researchers through the Chinese Virtual Herbarium (CVH, https://www.cvh.ac.cn/, accessed on 20 May 2024) platform. The digitization and data sharing of global plant specimens provide great convenience for taxonomists in obtaining detailed information on specimens and performing plant diversity work. The complete information on plant specimens is crucial for species classification, distribution, and system evolution. The information they contain holds significant research value [4]. Relevant quantitative analyses of whether the specimens stored in these databases can meet the basic needs of scholars, and the problems with their use exist for only some groups and for specific herbaria.

Bryophytes (including liverworts, mosses, and hornworts) are important components of biodiversity, with an estimated 17,900 species worldwide [5] and approximately 3500 species in China [6]. These plants play important roles in water conservation, plant colonization, seed germination, seedling growth, and forest regeneration [7]. In terms of species or evolution, liverworts are inferior to mosses but are richer than hornworts [8]. The evolutionary position of liverworts is the connection of lower and higher plant groups. They are not fragile plant groups in terms of environmental adaptability or plant species composition. Moreover, they play an irreplaceable and important role in the diversity of bryophytes and even plant species and the evolutionary history of the whole biosphere [9]. According to the latest data released by the Catalogue of Life China [10] (Species 2000, http://www.sp2000.org.cn/, accessed on 31 July 2024), there are 1191 species of liverworts belonging to 176 genera and 63 families in China. Compared with vascular plants and other groups, liverworts are often overlooked, and their diversity is underestimated in the surveyed areas because of a lack of researchers. As a result, the protection of liverwort diversity has become a weak point in overall biodiversity conservation efforts. With the continuous digitization of relevant literature and herbaria, summarizing the scattered multisource subset information is highly valuable. The establishment of species distribution datasets can be used as the main decision-making tool to describe the complete spatial coverage of species, select priority conservation areas, and predict species richness and potential distribution location models [11]. Based on the geographical distribution data of species specimens and related environmental variables, a species distribution model is used to simulate the potential distribution areas of species [12], which can be used to study the distribution patterns of large-scale biodiversity and the impacts of environmental changes on biodiversity [13]. For liverworts, which have not been fully explored and hold potential value, but are challenging to study, research on their collection specimens and geographical distribution is scarce. In this study, we analyzed the collection status and collection integrity of liverworts in China by utilizing the collection information of digital specimens of liverworts in China and taking provincial administrative divisions as the basic unit. Simultaneously, we predicted the potential geographical distribution areas of liverworts in China, in combination with climate data and geographical distribution information. We also clarified the main environmental factors affecting their distribution. These findings provide a theoretical basis for formulating countermeasures for the protection of liverwort diversity and further managing and utilizing plant resources.

## 2. Results

### 2.1. Basic Characteristics of Liverwort Specimens in China

A total of 55,427 specimens of liverworts in China were collected. After the records of liverworts identified to the genus level were excluded, the remaining 48,924 valid specimens were obtained. By verifying the distribution points and accurately recording specimen data at the provincial level, the remaining 48,840 specimen records were confirmed. The sources of the specimen data can be roughly divided into four categories (Figure 1a). A total of 3488 liverwort specimens were collected from the Flora Bryophytorum Sinicorum, accounting for 7.14% of the total number of specimens. Additionally, a total of 34,789 liverwort specimens were collected from the literature, accounting for 71.23% of the total specimens. Furthermore, a total of 1945 liverwort specimens were collected from a scientific investigation report of the reserve, accounting for 3.98% of the total specimens. Finally, a total of 8618 liverwort specimens were collected from the online databases, accounting for 17.65% of the total specimens. The majority of specimens were collected from 51 herbaria, with collection volumes ranging from 1 to 1176. Although some specimen information may be incomplete, resulting in lower collection volumes and units, this still provides a general overview of the status of liverwort specimen collections in China (Appendix A).

In this study, a total of 1300 species of liverworts belonging to 176 genera and 63 families in China were collected from the liverwort database based on specimens and literature data. However, the recorded number of liverwort specimens was only 1212; these specimens belong to 169 genera and 63 families. Only seven genera were not identified, including *Isopaches* [14], *Schljakovianthus* [15], *Konstantinovia* [16], *Cryptocoleopsis* [17], *Sinomylia* [18], *Dactylophorella* [19], and *Frullanoides* [20]. At the family level (Appendix A), *Lejeuneaceae* (8459 specimens), *Porellaceae* (4489 specimens), and *Plagiochilaceae* (3713 specimens) had the highest number of specimens. But *Pelliaceae* (152 specimens), *Dumortieraceae* (385 specimens), and *Conocephalaceae* (398 specimens) had the highest average number of specimens. There were 50 families with more than 10 specimens, accounting for 79.37% of all families. And 44 families had an average number of more than 10 specimens, accounting for 69.84% of all families. At the genus level (Appendix A), *Porella* (4370 specimens), *Frullania* (4155 specimens), and *Horikawaella* (3476 specimens) had the highest number of specimens. But *Reboulia* (643 specimens), *Barbilophozia* (671 specimens), and *Horikawaella* (1738 specimens) had the highest average number of specimens. There were 117 genera, with more than 10 specimens, accounting accounts for 66.48% of all the genera. There were a total of 103 genera, each with an average number of more than 10 specimens, accounting for 58.52% of all the genera. At the species level (Figure 1b), the average number of specimens collected per species was 40, which differed significantly from the actual collection number for each species. For example, only one specimen of *Frullania hainanensis* was collected, whereas 935 specimens of *Frullania muscicola* were collected. A total of 51.77% of the species were represented by fewer than 10 specimens, with 10.62% of these species represented by only 1 specimen. Only 9.7% of the species were represented by more than 100 specimens, with 0.61% of these species represented by more than 500 specimens.

Combined with the research history of bryophytes in China, the collection years of liverwort specimens were tallied, and the collection time period was categorized into four stages (Figure 2a). The first stage was prior to 1949, which was dominated by foreign collectors, and the collection amount was the lowest, accounting for only 1.93% of the total. The second stage, which spans 1950 to 1979, resulted in significant advancements in bryophyte research in China. Domestic scholars conducted large-scale studies on bryophytes and published the first monograph on the subject in China. This work accounted for 8.81% of the total collection during this period. The third stage, spanning from 1980 to 1999, significantly increased the number of investigations and studies on bryophytes across various regions of China. This led to the publication of local flora containing liverworts in numerous provinces and cities. These publications accounted for 12.96% of the total collection. The fourth stage spans 2000 to 2023. Some Chinese research teams have carried out some taxonomic revisions at the national or wider level of families and genera, and completed the construction of Chinese biological data network platforms such as Species 2000, NSII, and FOC. During this period, the amount of collection was the greatest, accounting for 76.29% of all the specimens collected (Figure 2b). The statistical results of the collected specimens with image records reveal that the collection amount also shows four obvious peaks. The results revealed that the collection of specimens with images was concentrated mainly from 2010 to 2023, and the collection amount accounted for 36.56% of the total number of image specimens, whereas the collection amount from 1940 to 1949 was the lowest, accounting for only 0.58% of the total.

The statistical results of liverwort specimen data in China’s provincial-level administrative regions indicate that, with the exception of Tianjin, there are records of liverwort specimens in all provincial-level administrative regions in China. However, there has been a significant imbalance in the collection of specimen data (Figure 3). From the perspective of the number of specimens in each province, more than 2000 specimens were collected in six provinces, accounting for 58% of the total number of liverwort specimens in China, of which the most were collected in Guizhou Province (10,488 specimens) and Yunnan Province (8259 specimens). The number of specimens collected ranged from 1000 to 1999. These specimens are distributed across 11 provinces, accounting for 30.89% of the total number of liverworts in China. Additionally, there are 11 provinces where the number of collected specimens falls within the range of 200–999, representing 10.72% of the total number of liverworts in China. Furthermore, there are six provinces in which fewer than 200 specimens have been collected, accounting for only 0.47% of the total number of liverworts in China.

### 2.2. Analysis of the Completeness and Sufficiency of the Liverwort Collection in China

From the perspective of the whole region of China (Figure 4), the collection sites of liverwort specimens are mainly concentrated in the Greater Khingan Mountains and Lesser Khingan Mountains in Northeast China, as well as the Changbai Mountains. The collection amount of liverworts from the southern Qinling Mountains is relatively high. The collection areas on the west side are mainly the Yinshan Mountains and Helan Mountains. The specimen collection sites in the northwestern region are concentrated in the Tianshan Mountains, Qilian Mountains, and southern Himalayas. The collection sites in the southwest region are concentrated in the Hengduan Mountains and Yunnan–Kweichow regions. Moist tropical and subtropical regions are hotspots for liverwort diversity. The species richness, specimen collection, and sampling sites are all in the Hengduan Mountains and Yunnan–Guizhou Plateau, which are far greater than those in other regions. However, the observation values and specimen collection rates of liverworts in certain provinces and regions of Central China and the North China Plain are relatively low. Specifically, the species richness and number of specimens collected in five provincial regions, including Ningxia, Shanxi, and Shanghai, are less than 100.

When the collection integrity of each family was evaluated using the ratio method, it was found that the Chao1 value could not be calculated because the data of species specimens from 46 families, represented by *Haplomitriaceae*, did not meet the calculation conditions. The ratios of the remaining seventeen families were greater than −0.3, with eleven families such as *Aytoniaceae*, *Radulaceae*, and *Cephaloziaceae* having ratios greater than −0.1, indicating that they were complete families. The remaining six families, including *Marchantiaceae*, *Anastrophyllaceae*, and *Plagiochilaceae*, are considered to be relatively comprehensive (Appendix A). Only twenty-three genera met the calculation conditions. Among them, eighteen genera, such as *Frullania*, *Radula*, and *Plagiochila,* were collected as complete genera. The remaining five genera, including *Mannia*, *Marchantia*, and *Solenostoma,* were collected as relatively complete genera (Appendix A). Notably, the Chao1 estimators of ten families and nine genera, such as *Solenostomataceae*, *Solenostoma*, *Plagiochilaceae* and *Plagiochila*, exceed the numbers recorded in the literature, whereas the estimators of the remaining families and genera are less than the numbers recorded in the literature.

### 2.3. Impact of Climatic Factors on the Geographical Distribution of Liverworts in China

The test results revealed that the area under the ROC curve was 0.872 after the MaxEnt model was employed to cross-validate the distribution information and environmental factors of liverworts for 10 iterations and taking the mean value. These findings indicate that the model exhibited strong predictive ability and can be used to forecast the potential distribution area of liverworts in China. According to the results of the knife-cut method in the MaxEnt model, the regularized training gain of nine key environmental variables (Appendix A) is obtained, and then combined with the estimated value of the relative contribution of the MaxEnt model to the nine environmental variables and the value of the importance of the arrangement (Appendix A). The cumulative contribution rates of the four environmental factors, including the precipitation of the warmest quarter (Bio18), the annual temperature range (Bio7), the mean temperature of the warmest quarter (Bio10), and the precipitation of the driest month (Bio14), reached 83.2%. Additionally, the cumulative replacement importance value reached 73.3%, and the regularized training scores were relatively high. The normalized training score of elevation (Elev) was relatively low, but the contribution rate and replacement importance were high, indicating its significance in the analysis. Therefore, it was deemed necessary to retain this variable. The five environmental variables Bio18, Bio7, Elev, Bio10, and Bio14 were identified as the primary factors influencing the distribution of liverworts in China.

The single-factor response curves of the aforementioned five main environmental factors and the presence probabilities of the liverworts are depicted in Figure 5. It is widely accepted that when the probability of liverwort existence exceeds 0.5, the corresponding environmental factor value is deemed suitable for liverwort growth. The probability of the presence of liverworts in China increased with increasing precipitation in the warmest quarter (Bio18), peaked at 538.9 mm, and then stabilized after a decline. The annual temperature range (Bio7) was negatively correlated with the survival probability of liverworts. Specifically, when the annual temperature ranged from 5.0 °C to 30.4 °C, the survival probability of liverworts was greater than 0.5. The elevation (Elev) and the mean temperature of the warmest quarter (Bio10) exhibited a multipeak relationship with the survival probability of liverworts. Optimal survival peaks were observed when the elevation ranged from 2083.96 to 2187.3 m, and when the mean temperature of the warmest quarter fell between 23.4 °C and 24.8 °C and between 28.3 °C and 28.5 °C. When the precipitation of the driest month (Bio14) exceeded 8.65 mm, the survival probability of liverworts was greater than 0.5. Furthermore, when the precipitation of the driest month reached 173.28 mm, the probability of occurrence of liverworts peaked and then remained unchanged thereafter.

According to the classification criteria for suitable areas of liverworts in China, we obtained a potential distribution map of liverworts (Figure 6), which was highly consistent with the actual sampling points of liverworts in China. The high-suitability area of liverworts in China is 133.57 million km^2^. The high-suitability areas are distributed mainly south of the Yangtze River, extending from the central Yunnan–Guizhou region to the Qinling Mountains, and are concentrated in the Hengduan Mountains and the southern end of the Himalayas in the northwest direction. The southern region is concentrated in the Wuyi Mountains, Nanling Mountains, and coastal areas, and the northeastern region is distributed mainly in the Changbai Mountains and Lesser Xing’an Mountains. The middle- and low-suitability areas are centered around the high-suitability area and gradually extend to the surrounding areas, whereas the remaining areas are unsuitable areas.

## 3. Discussion

### 3.1. Analysis of the History and Current Situation of Liverwort Specimens in China

During the study of liverwort taxonomy in China, the collection of liverworts exhibited four distinct peaks. The overall collection curve of liverworts indicated that the periods from 2010 to 2023 (30,260 specimens, 56.82%), 2000 to 2009 (10,369 specimens, 19.47%), and 1980 to 1989 (4255 specimens, 7.99%) were the most abundant times for liverwort collection, coinciding with the historical stage of bryophyte taxonomy research in China. The study of bryophytes in China began relatively late. From 1845 to 1936, several European botanists primarily collected bryophyte specimens in various regions of China. It was not until the 1940s that local scholars in China truly started studying bryophytes, and a comprehensive investigation of bryophytes was subsequently conducted on a large scale in the country. The collection sites include Anhui, Hainan, Guangdong, Yunnan, Fujian, and other provinces and regions [21]. Vigorous bryophyte specimen collection and collection activities have been vigorously carried out, and a number of bryophyte herbaria have been established, gradually increasing the current scale and collection status. In the following decades, as investigations and research became increasingly extensive, a number of bryophyte genera and local lists involving liverworts in various regions of China were successively published [22]. In the past ten years, with the taxonomic research beginning in the era of molecular biology, some national research teams have successively carried out the taxonomic revision of some families and genera of liverworts nationwide or within a wider range, and identified many problems in the research, necessitating the continued collection of specimens. Moreover, in molecular biology research, voucher specimens need to be collected. Therefore, the rate of specimen collection has gradually recovered, and the amount of collection has increased annually. The combination of the collection history of liverworts provides a verification direction for the taxonomic research process of liverworts in China.

Statistical analysis revealed imbalances in the distribution of liverwort data among different groups, collection units, and geographical areas. The majority of the specimens were obtained from 51 herbaria, including the Herbarium of the Institute of Botany, Chinese Academy of Sciences, Beijing, China.(PE), the Herbarium of Kunming Institute of Botany, Chinese Academy of Sciences, Yunnan, China(KUN), and the Herbarium of the Natural History Museum of Guizhou University, Guizhou, China(GACP). Guizhou had the greatest number of liverwort specimens at 10,488, followed by Yunnan with 8259 specimens, Hainan with 3887 specimens, and Shaanxi with 3527 specimens. These four regions account for 47.20% of all liverworts in China. The distribution was mainly concentrated in the Hengduan Mountains, Yunnan–Guizhou Plateau, Qinling Mountains, and most coastal areas. To date, research teams have conducted studies on liverworts in various periods in China’s biologically diverse regions and have amassed a substantial collection of liverwort research materials or voucher specimens. However, detailed taxonomic research on liverworts has not yet been carried out in many regions of the country. Secondly, the collection of liverwort specimens is highly professional. There are currently no herbaria or dedicated teaching or scientific research units specializing in liverwort research. Generally, liverwort specimens are either not preserved at all or rarely preserved. Furthermore, the liverworts collected in the specimen collections show obvious preference and incompleteness at the family and genus levels, such as *Conocephalaceae*, *Frullaniaceae*, *Porellaceae*, and *Lejeuneaceae*. These biases may be attributed to the wide distribution of these liverwort groups and could also be strongly correlated with the groups studied by experts and local resources.

Currently, the digital specimen information of liverworts is incomplete. First, many units have not sorted out their overstocked specimens in a timely manner, leading to missing collection information recorded by the specimens. Second, owing to the establishment and classification changes of many liverwort species, the specimen information has not been updated promptly. A comparison between the family and genus information recorded in the literature and specimen records reveals significant discrepancies between the two. Third, some liverwort specimens have not been digitized and shared, which is an essential step in the process of specimen digitization. As a result, a significant amount of liverwort specimen data can currently be obtained only through a literature review. However, the lack of specimen information and limited records of liverwort species in certain areas have significantly impeded research efforts and greatly restricted the protection, development, and utilization of liverwort plants.

### 3.2. Analysis of the Completeness of the Collection of Liverworts in China

According to the evaluation results of the collection group deviation, there are nineteen families and seventy-three genera of liverworts with an average number of specimens < 10, accounting for 30.16% and 41.47% of the total families and genera of liverworts in China, indicating that the collection of some families and genera is insufficient, and that the collection is mainly concentrated in common families and genera as well as large families and genera with a wide distribution range. Among the 17 families and 23 genera of liverworts for which the Chao values can be calculated, the proportion of families with complete collections reached 64.71%, and the proportion of genera with complete collections reached 78.26%. There are only six families and five genera of incompletely collected liverworts, such as *Solenostomataceae*, *Solenostoma*, *Porellaceae,* and *Porella*, and there is a relationship between incompletely collected families and genera. The ratio method is more sensitive to the number of rare species. Therefore, when the number of specimens collected in a family is too small or the number of collections is inconsistent, the Chao value may not be calculated. However, it should be noted that this does not necessarily indicate an incomplete collection of these groups. The calculated collection integrity serves as a basis for judgment and requires an overall comprehensive evaluation.

Combined with the collection status of provincial areas in China, these findings also indicate that the collection integrity of some areas and groups is high, and that investigations of liverworts are relatively comprehensive. For the areas and groups where investigations and research on liverworts are lacking, it is necessary to focus on completing the collection of families and genera as well as increasing efforts in collection, preservation, and identification in areas with insufficient coverage. As further investigations into liverworts are conducted in the future, more groups will likely be discovered. Although this paper focuses on provincial administrative regions as its research unit, resulting in lower accuracy and somewhat rough conclusions, it can still provide valuable macro directions and suggestions for future collections of liverwort specimens.

### 3.3. Simulation Analysis of Suitable Areas for Liverworts in China

In this study, we utilized the MaxEnt model to investigate the potential spatial distribution of liverworts in China. The MaxEnt model stands out from other methods because of its ability to incorporate spatial deviation data and limited species occurrence records into the modeling process [23]. In addition, only existing points can be used for modeling analysis, and a built-in knife-edge test can be performed, which allows the importance of a single environmental variable to be estimated when calculating species distributions [24]. The model used in this study demonstrated a strong predictive effect on the potential distribution area of liverworts in China, leading to a high AUC value.

The growth and distribution of bryophytes in China are influenced by various environmental conditions, with the environment playing a crucial role in their diversity and distribution. Among the environmental factors affecting the geographical distribution of liverworts, temperature-related variables (Bio10, Bio9, Bio7, Bio3, and Bio2) contributed 52.6%, precipitation-related variables (Bio18, Bio15, and Bio14) contributed 39%, and terrain-related variables (Elev) contributed 8.6%. Therefore, the primary environmental variables influencing the potential distribution of liverworts in China are temperature and precipitation, followed by terrain variables. The main environmental factor affecting the distribution of liverworts is precipitation in the warmest quarter, followed by the annual temperature range. Various scholars have obtained similar results when studying the primary environmental factors influencing the distribution of bryophytes. Relevant studies have indicated that precipitation, temperature, and altitude can impact the species and distribution of bryophytes [25]. The presence of water plays a critical role in the distribution of bryophytes, with bryophytes being more sensitive to precipitation than to temperature, as supported by previous research [26,27]. In contrast, bryophytes are tolerant of warmer habitats and are therefore not limited to cooler mountainous regions. They are able to inhabit niches with significant temperature fluctuations, which may also experience more pronounced changes in precipitation [28].

Based on the prediction results, liverworts are found to have broad suitability areas in the tropical and subtropical regions of China. The high-suitability areas for potential distribution are located primarily in Southwest China and South China, indicating a patchy distribution pattern. On the one hand, these findings confirm the high species richness of liverworts in the southern region, and also prove the correlations between the predicted environmental factors and the climate and topography of the suitable area. This study speculates that it is precisely because of the special biological characteristics of dampness and heat resistance that the southern part of China, which has sufficient precipitation and a suitable temperature, is a high-suitability area for liverworts. Studies have indicated that the diversity of liverworts in many arid regions is limited, suggesting a high reliance on readily available water such as precipitation or clouds. Only a small number of specialized taxa have been able to reduce this dependence on water and therefore can thrive in dry areas [29]. This also explains the phenomenon of large-scale unsuitable areas in Qinghai, Xinjiang, and western Tibet. Owing to limited data on the detailed local distributions of individual species, particularly in the tropics, identifying priority areas for bryophyte diversity appears to be the most feasible method [30]. Research on the impact of climate change on the potential distribution area of liverworts holds significant theoretical importance for the future protection and rational development of liverwort resources in China.

## 4. Materials and Methods

### 4.1. Data Source

The data sources of this study are the following: (1) Flora books. These books mainly include ‘*Bryophytorum Sinicorum*’ (Volumes 9 and 10) [31,32]. (2) Journal papers. By July 2024, a total of 578 papers had been published in the China Knowledge Network and Web of Science databases, as well as in the professional bryophyte journals ‘*Bryologist*’ and ‘*Journal of Bryology*’, including journal papers and dissertations. (3) List of protected plants. A total of 32 scientific expedition reports from various regions of China have been compiled. (4) Online databases. These databases include the National Specimen Information Infrastructure [33] (NSII, http://www.nsii.org.cn/2017/home.php, accessed on 30 May 2024), the Global Biodiversity Information Facility [34] (GBIF, https://www.gbif.org/.cn/, accessed on 30 June 2024), and the Missouri Botanical Garden [35] (Tropicos, https://www.tropicos.org/, accessed on 15 July 2024). The collection person, collection time, specimen museum, collection number, and origin information of the specimen data were recorded, and statistical analysis was performed using Excel. The full name and code information of each herbarium are derived from the herbarium of the New York Botanical Garden [36] (NYBG, http://sweetgum.nybg.org/science/ih/, accessed on 23 June 2024) (Appendix A).

### 4.2. Data Standardization

As plant taxonomy research has advanced, the use of scientific names for plants has evolved. Additionally, different collectors may employ varying recording styles and levels of completeness in documenting specimen collection information, leading to incomplete or even ambiguous data associated with existing specimens. Therefore, this study standardized the obtained specimen data, and the specific methods are as follows: (1) The Taxonomic Name Resolution Service v4.1 [37] (TNRS, http://tnrs.iplantcollaborative.org, accessed on 15 July 2024) was used to correct spelling errors in the scientific names of the specimen records. However, the species view of TNRS is not adopted, spellings that TNRS could not solve were corrected using Tropicos. (2) Doubts regarding specimen species identification, errors in geographical distribution information, missing collection sites, and collection records outside the administrative region of China were discarded. (3) All the specimen information collected from Chinese liverworts is based on the provincial (e.g., municipalities and autonomous regions) administrative regions as the basic unit of data. For the data of incomplete records of collection sites and changed place names, the Baidu map was used to query the place name fields recorded on the specimens, complete the data as much as possible, and complete the verification of place names at the same time. After the completion of the data standard, the Baidu map coordinate picking system [38] (https://api.map.baidu.com/lbsapi/getpoint/index.html, accessed on 30 July 2024) was utilized to retrieve latitude and longitude information corresponding to the specimen collection points. The statistics indicate that a total of 8957 distribution points for liverworts in China were obtained, with the elimination of distribution points gathered within a 5 km × 5 km grid. Ultimately, 729 liverwort distribution points in China were identified, and ArcGIS 10.4 was employed to create a geographical distribution map.

### 4.3. Evaluation of Collection Integrity and Sampling Adequacy

The ratio method is one of the commonly used methods in the field of biodiversity. The Chao1 estimator was first proposed by Chao in 1984. It is assumed that if a random sample is taken in a group, when the number of species collected is only 1 but it is still continuously discovered, it indicates that there are still species that have not been discovered; until the number of collections of all species is at least 2, it indicates that no new species will be found. The formula is Chao1 = S_obs_ + F_1_^2^/2F_2_, where S_obs_ represents the number of species observed in the sample, F_1_ represents the number of species with only one individual, and F_2_ represents the number of species with only two individuals. In this work, the ratio of the difference between the number of species actually recorded and the Chao1 estimator to the Chao1 estimator is used to evaluate the integrity of the specimen collection of each family and genus of liverworts, that is, the collection integrity = (observed value-theoretical value)/theoretical value. The smaller the ratio, the more incomplete the collection [39]. In this study, the ratio was limited to <−0.3 for an incomplete collection, [−0.3, −0.1] for a relatively complete collection, and >−0.1 for a complete collection.

Identifying individuals of bryophyte specimens seems to be a difficult problem. However, in line with the IUCN SPSC recommendation to ‘assume an average area occupied by a mature individual and estimate the number of mature individuals from the area covered by the taxon’, the Swedish Red List Committee for Bryophytes interpreted ‘mature individuals’ in a pragmatic way by using the concept of an ‘individual-equivalent’. Terricolous taxa growing on the ground on various substrates (i.e., sand, gravel, earth, and litter), or saxicolous taxa growing on cliffs or on other more or less flat surfaces represent an ‘individual-equivalent’ = 1 m^2^ [i.e., 1 m^2^ in which the taxon occurs, whether as a single ramet or as a dense carpet of many ramets covering most of the surface]. Saxicolous or terricolous taxa on boulders (the latter, for example, in earth-filled fissures) represent ‘an individual equivalent’ = 1 boulder on which the taxon is growing. Epiphytic and epiphyllic taxa represent ‘an individual-equivalent’ = 1 tree or 1 shrub on which the taxon is growing. Epixylic taxa represent ‘an individual-equivalent’ = 1 log on which the taxon is growing. This is almost the same as the process of collecting bryophytes, so that each specimen of liverworts collected is studied as an individual [40].

### 4.4. Climatic Factor Variable Data Screening

One topographic and nineteen environmental factor variable data points with a resolution of 2.5 arc-minutes were downloaded from the global climate data website (http://www.worldclim.org/, accessed on 15 May 2024), and the obtained environmental change data were exported in asc format in ArcGIS according to the Chinese boundary. To avoid overfitting of the model caused by the high collinearity among the environmental factors [41], the knife-cut method in the MaxEnt model was used to analyze the contribution rates of 20 postclipping environmental factors to the model prediction. First, 729 liverwort specimens and 20 environmental variables were loaded into the MaxEnt model to obtain the contribution rates of 20 environmental variables. Then, the values of 729 liverwort specimens and 20 environmental variables were extracted from each sample point in ArcGIS, and the data corresponding to each environmental variable were obtained. Pearson correlation analysis was performed on the data using SPSS Statistics 27 [42]. When the absolute value of the correlation coefficient was greater than 0.85, the environmental factors with small relative contribution rates to the model prediction were eliminated. Finally, nine key environmental variables were selected for the final niche simulation, and the main environmental factors affecting the distribution of liverworts were analyzed (Table 1). The map data source is the Chinese standard map provided by the standard map service website of the National Bureau of Surveying, Mapping and Geoinformation (http://www.mnr.gov.cn/, accessed on 10 May 2024).

### 4.5. MaxEnt Model Analysis of Liverworts in China 

The latitude and longitude information and environmental layers of the distribution points of the Chinese liverwort specimen data were imported into MaxEnt 3.4.4 [43]. In the model analysis, the maximum number of iterations was set to 10,000. Bootstrapping was repeated 10 times, and the response curves and jackknife test were selected to analyze the environmental variables affecting the distribution of liverworts. MaxEnt software automatically drew the ROC curve and calculated the area under the curve (AUC) (area under the receiver operating characteristic curve) to test the accuracy of the model prediction [13]. Referring to the evaluation index of Hanley and Mcneil [44], the AUC value was used as a measure of model prediction: AUC values were between 0.5 and 0.6 (prediction failure), 0.6 and 0.7 (poor prediction effect), 0.7 and 0.8 (general prediction effect), 0.8 and 0.9 (good prediction effect), and 0.9 and 1.0 (very good prediction effect). The AUC of the model fitting in this paper was >0.8.

The environmental data and liverwort distribution point data selected were input into the constructed MaxEnt model. The average value of the model after 10 runs in asc format was then loaded into ArcGIS 10.4. The reclassification tool was used to categorize suitable areas for Chinese liverworts into four grades using natural breaks, and the area of each grade was subsequently calculated. The critical value for suitable and unsuitable areas of liverworts in China was determined to be 0.114. This was followed by categorizing the areas into unsuitable (0–0.114), low-suitability (0.114–0.300), medium-suitability (0.300–0.502), and high-suitability (0.502–0.973).

## 5. Conclusions

This study describes the current situation and shortcomings of liverwort specimen collection in China and provides basic information for the rational conservation of liverworts in China in the future through important species distribution data. There is a need for a more extensive inventory and investigation of key and potential distribution areas of liverworts, especially in underdeveloped ecoregions. It is highly recommended to digitize and share specimens from local herbaria, museums, universities, botanical gardens, institutions, and private collectors. International research collaborations are also recommended to expand and strengthen specimen digitization. These collaborative data will make the research and conservation of bryophyte resources more representative and comprehensive and provide the basis for the national inventory necessary for national strategic biodiversity and action plans.

## Figures and Tables

**Figure 1 plants-13-02583-f001:**
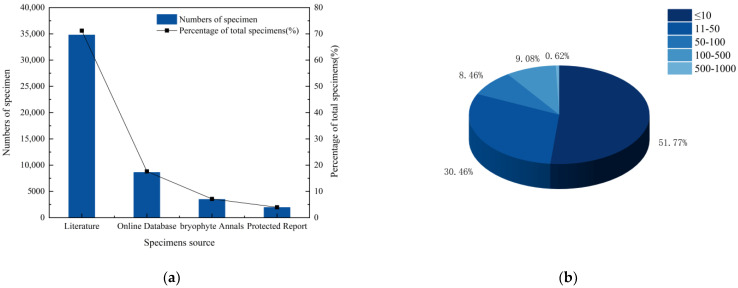
Statistical depiction of liverwort specimen data collected in China. (**a**) The source of collections of liverwort specimens in China. (**b**) Status of liverwort specimens in China at the species level.

**Figure 2 plants-13-02583-f002:**
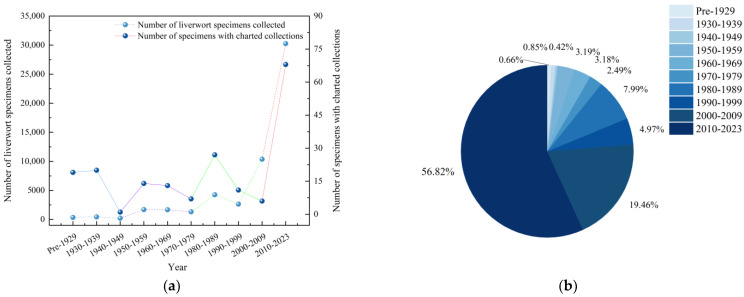
Collection period of liverwort specimens in China. (**a**) Year of collection of liverwort specimens in China. (**b**) Relative proportions of liverwort specimens collected at various periods in China.

**Figure 3 plants-13-02583-f003:**
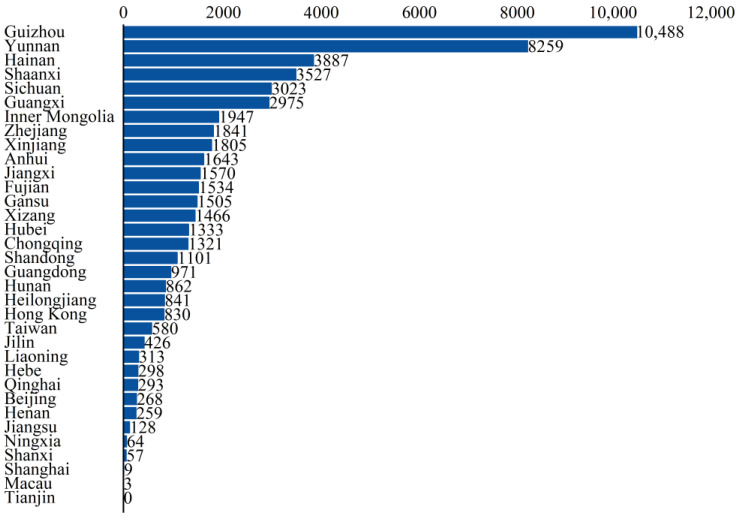
The number of liverwort specimens from provincial regions of China.

**Figure 4 plants-13-02583-f004:**
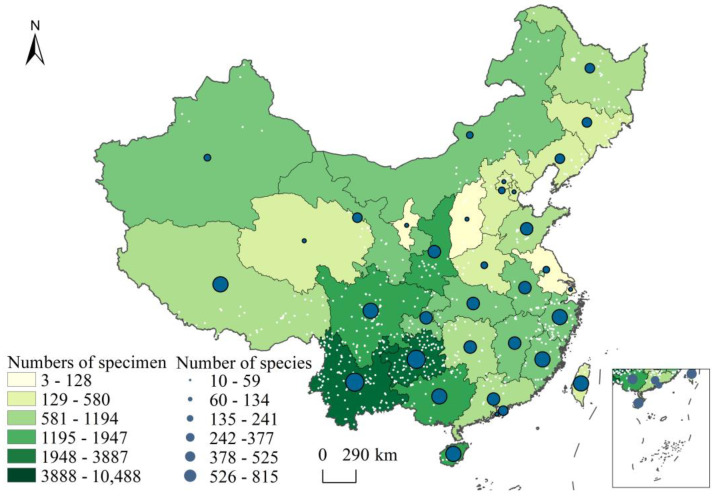
Distribution of liverwort specimens and species in different provinces of China. The white dots indicate sampling points with duplicate sites removed.

**Figure 5 plants-13-02583-f005:**
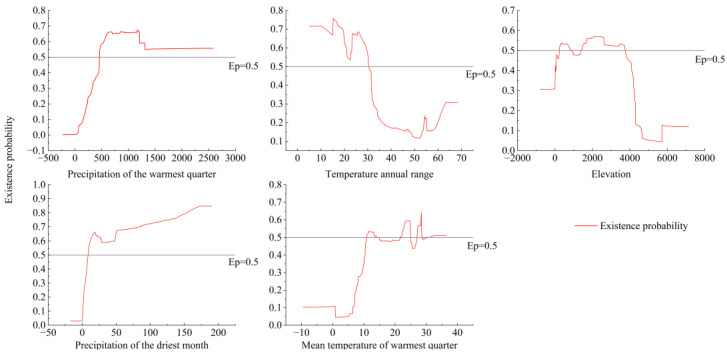
Response curves of dominant environmental variables.

**Figure 6 plants-13-02583-f006:**
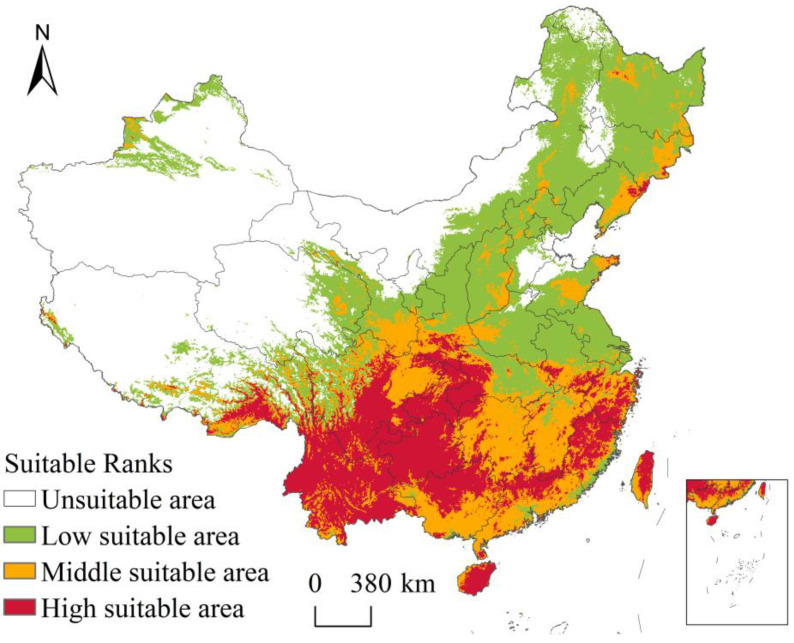
Simulation results of the MaxEnt model of suitable areas for liverworts in China.

**Table 1 plants-13-02583-t001:** Contribution percent of each environmental variable in MaxEnt modeling.

Variable	Environmental Variable	Unit
Bio2	Average daily difference in temperature	°C
Bio3	Isothermal	-
Bio7	Temperature annual range	°C
Bio9	Mean temperature of driest quarter	°C
Bio10	Mean temperature of warmest quarter	°C
Bio14	Precipitation of the driest month	mm
Bio15	Precipitation seasonality (coefficient of variation)	%
Bio18	Precipitation of the warmest quarter	mm
Elev	Elevation	m

## Data Availability

The raw data that support the findings of this study are from public databases, such as the GBIF (https://www.gbif.org/, accessed on 30 June 2024), the Missouri Botanical Garden (https://tropicos.org, accessed on 15 July 2024) and NSII (http://www.nsii.org.cn/2017/home.php, accessed on 30 May 2024). The liverwort papers are from China National Knowledge Infrastructure and Web of Science.

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
