# Peer review of "Current Status of Liverwort Herbaria Specimens and Geographical Distribution in China"

_plants, 2024, doi:10.3390/plants13182583_

Round 1
Reviewer 1 Report
Comments and Suggestions for Authors
The submitted manuscript is well-prepared, with results that are clearly presented and effectively illustrated through appropriate methods. As a curator of herbarium collections, I recognize the significance of this work for systematic bryology.
While the study may not represent a groundbreaking innovation in the field, I recommend its publication due to its potential for substantial practical application.
However, I suggest enlarging the captions in the figures. Currently, they are significantly smaller than the main text, making them difficult to read.
Author Response
Comments 1:[I suggest enlarging the captions in the figures. Currently, they are significantly smaller than the main text, making them difficult to read.]
Response 1: [Thank you for pointing this out. I agree with this comment. Therefore, I will revise it immediately.]
Reviewer 2 Report
Comments and Suggestions for Authors
I am not a native speaker, but I usually have no difficulty reading articles in English. However, I do not understand many of the suggestions in this article. It seems that the authors are expressing their thoughts incorrectly, or translating them incorrectly.
Starting with the title, many phrases are difficult to understand. It seems that the authors use the word "collect" incorrectly. Some ph It seems that something has not been completed, or incorrectly translated. Some phrases seem unfinished, or as if a word is missing in them. The most important thing is that it is unclear what did the authors want to say in specific phrases.

Comments on the Quality of English Language
The authors have done a lot of work on the study of sources that provide data on the liverworts of China. However, they were unable to present the results correctly and clearly. As presented, the article is very difficult to read and many of the provisions are simply incomprehensible.
The title of the article is somewhat ambiguous and does not reflect the content of the paper. What do the authors mean by “Current status of liverwort herbarium specimens”? Geographical distribution of what do the authors mean? Specimens? Herbariums? Liverworts?
Then in the abstract, what did the authors mean by “The average number of liverwort specimens collected was 40”. The average number of where or collected by whom? Etc. Etc.
There is no explanation for some terms (for example, individual) used by the authors. One individual in liverworts, which often reproduce vegetatively and form clones, (see Bergamini, A., Bisang, I., Hodgetts, N., Lockhart, N., van Rooy, J. and Hallingback, T. (2019). Recommendations for the use of critical terms when applying IUCN redlisting criteria to bryophytes. Lindbergia. doi: 10.25227/linbg.01117).
Some notes are as well in the text, but in general the article needs to be carefully thought out and redone.
Author Response
Comments 1: [40 samples where? in the one particular herbarium? Or particular family?]
Response 1: [Thank you for your comment. I'm sorry for the ambiguity .The 40 specimens are the calculated average specimen of liverworts. I wanted to use the overall average specimen count to reflect the current state of liverwort collection, which has been expressed more explicitly in the abstract.]
Comments 2: [the stages of what are meant?]
Response 2: [Thank you for your comment. Lines 130 to 149 of the article divide the collection time of Chinese liverworts into four stages. I also think that there is some ambiguity here, and rephrase it.]
Comments 3: [There is no way to obtain morphological information for small bryophytes during digitalization of samples.]
Response 3: [Thank you for your comment. I agree that morphological information cannot be obtained in bryophyte specimens, so I will correct this error.]
Comments 4: [Do you mean specimens?
What is meant by "were collected in the Flora?"What is meant by flora with a capital letter? It's not clear to me.]
Response 4:[Thank you for your comment. I'm sorry for the translation issues, A total of 3,488 liverwort specimens were collected from the Flora Bryophytorum Sinicorum(Book). In line 345 of the text, changes have been made.
Comments 5: [Do you mean they were found in literature sources? I've never seen such an expression.]
Response 5:[Thank you for your comment. The liverwort database was formed based on the literature of bryophytes from 1958 to 2024, which included 1304 species of liverworts belonging to 176 genera and 63 families. In lines 358 to 360 of the text, it has been restated.]
Comments 6: [Firstly, there should be a comma before including, and secondly, it is unclear why the authors distinguish these genera separately.]
Response 6:[Thank you for your comment. A comma before inclusion has been added. These seven genera refer to the liverwort taxa for which no specimens have been found. It has been expressed in detail in the text.]
Comments 7: [bryophytes, not mosses only.]
Response 7:[Thank you for your comment. I agree with your comment, which has been corrected in the text.]
Comments 8: [What do you mean by individual?See Bergamini, A., Bisang, I., Hodgetts, N., Lockhart, N., van Rooy, J. and Hallingback, T. (2019). Recommendations for the use of critical terms when applying IUCN redlisting criteria to bryophytes. Lindbergia. doi: 10.25227/linbg.01117]
Response 8:[Thank you very much for your comments. Identifying individuals of bryophytes specimens seems to be a difficult problem. Your recommended article provides a very comprehensive answer to this question. This method is closely related to bryophyte specimens collection, and has been supplemented in lines 1702 to 1801 of the article Methodology.]
Comments 9: [The title of the article is somewhat ambiguous and does not reflect the content of the paper. What do the authors mean by “Current status of liverwort herbarium specimens”? Geographical distribution of what do the authors mean? Specimens? Herbariums? Liverworts?]
Response 9:[Thank you very much for your comments. The collection information of liverwort specimens in China was statistically analyzed, and the collection of liverworts was analyzed from the taxa, collection time and collection location to reflect the current situation of liverwort specimens in China. Further research was carried out based on the detailed geographical distribution information of the collected specimens of liverworts. This forms the title of the article.]
Comments 10: [The English is very difficult to understand/incomprehensible.]
Response 10:[Thank you for your comment. I agree that there are some errors in the English language . I will re-edit the English language of this article.]

Reviewer 3 Report
Comments and Suggestions for Authors
Overall, the writing needs to be tighter. The same thing is repeated several times. The figures are quite small, and it is difficult to read any writing on them. Please note that the plural of herbarium is herbaria.
Comments on the Quality of English Language
The English certainly needs work. It is often difficult to tell when the authors did something versus when they were reporting on someone else's data.
Author Response
Comments 1: [The figures are quite small, and it is difficult to read any writing on them. ]
Response 1: [Thank you for your comment. I agree with your comment. I will change it right away.]
Comments 2: [Please note that the plural of herbarium is herbaria.]
Response 2: [ Thank you for your comment. I agree with your comment. I will pay attention to the use of the herbarium and herbaria.]
Comments 3: [The English certainly needs work. ]
Response 3: [Thank you for your comment. I agree that there are some problems with the manuscript English. I'm sorry that this is happening, and I will re-edit the manuscript.]
Reviewer 4 Report
Comments and Suggestions for Authors
This paper documents the liverworts in Chinese herbaria. This is a useful compilation but does not reveal any surprising new information. As it stands the work is worth publishing but does not fit comfortably into a research-based journal. The paper would benefit from light editing of the English by around 10%.
The work is well organised .
Interest in the work would be increased if it included comparisons between moss herbaria in China and with liverwort herbaria in other countries. The concluding paragraph should be deleted as it does not contain any useful information.
Comments on the Quality of English Language
Would benefit from light editing to reduce the length by 10%
Author Response
Comments 1:[Interest in the work would be increased if it included comparisons between moss herbaria in China and with liverwort herbaria in other countries.]
Response 1: [Thank you very much for your comment, and I think this is a very good proposal, and I will focus on collecting information on bryophyte specimens from other countries in future research to make more meaningful discoveries.]
Comments 2:[The concluding paragraph should be deleted as it does not contain any useful information.]
Response 2: [Thank you for your comment. I will carefully consider the content of this last paragraph, which is intended to summarize the importance of specimens in botanical research, and may not adequately reflect the article.]
Comments 3: [Minor editing of English language required. Would benefit from light editing to reduce the length by 10%.]
Response 3: [Thank you for your comment. I agree that some English expressions are incorrect and some sentences are repetitive. I will re-edit the English language of this article.]
Round 2
Reviewer 4 Report
Comments and Suggestions for Authors
The revised version is a big improvement on the original and, in my view, is now suitable for publication though I would have liked to have seen the inclusion of information on other liverwort herbaria.
Author Response
Comments 1: [The revised version is a big improvement on the original and, in my view, is now suitable for publication though I would have liked to have seen the inclusion of information on other liverwort herbaria.]
Response 1:[Thank you for your comment. I agree with your comment. This paper mainly discusses the collection of liverwort specimens in China, but the collection of liverwort specimens in other countries herbaria are also very meaningful, and I will study it in the future.]